# Assessing the mechanisms of multi-drug resistant non-typhoidal *Salmonella* (NTS) serovars isolated from layer chicken farms in Nigeria

Idowu Oluwabunmi Fagbamila[1]*, Elena Ramon[2], Antonia A. Lettini[2], Maryam Muhammad[1], Alessandra Longo[2], Keti Antonello[2], Mabel K. Aworh[3], Jacob K. P. Kwaga[4], Paul A. Abdu[5], Jarleth U. Umoh[4], Junaidu A. Kabir[4], Antonia Ricci[2], Lisa Barco[2]

1 Bacterial Research Division, National Veterinary Research Institute, Vom, Plateau State, Nigeria, 2 WOAH and National Reference Laboratory for Salmonellosis, Istituto Zooprofilattico Sperimentale delle Venezie, Legnaro, PD, Italy, 3 Department of Population Health and Pathobiology, College of Veterinary Medicine, North Carolina State University, Raleigh, North Carolina, United States of America, 4 Department of Veterinary Public Health, Faculty of Veterinary Medicine, Ahmadu Bello University, Zaria, Nigeria, 5 Department of Veterinary Medicine, Faculty of Veterinary Medicine, Ahmadu Bello University, Zaria, Nigeria

* dridowu4u@yahoo.com

**Data Availability Statement:** All relevant data are within the paper and its Supporting Information files.

## Abstract

### Background

In Nigeria, there have been reports of widespread multiple antimicrobial resistance (AMR) amongst *Salmonella* isolated from poultry. To mitigate the impact of mortality associated with *Salmonella* on their farms, farmers resort to the use of antimicrobials without sound diagnostic advice. We conducted this study to describe the AMR patterns, mechanisms and genetic similarities within some *Salmonella* serovars isolated from different layer farms.

### Method

We determine the AMR profiles of two hundred *Salmonella* isolates, selected based on frequency, serovar, and geographical and sample type distribution. We also assessed the mechanisms of multi-drug resistance for specific genetic determinants by using PCR protocols and gene sequence analysis. Pulsed-field gel electrophoresis (PFGE) was conducted on seven selected serovars to determine their genetic relatedness.

### Results

Of 200 isolates, 97 (48.5%) revealed various AMR profiles, with the multiple antibiotic resistance (MAR) index ranging from 0.07–0.5. Resistance to ciprofloxacin was common in all the multi-drug resistant isolates, while all the isolates were susceptible to cefotaxime, ceftazidime, and meropenem. Genotypic characterization showed the presence of resistance genes as well as mutations in the nucleotide genes with subsequent amino acid substitutions. Fifteen isolates (43%) of *S*. Kentucky were indistinguishable, but were isolated from four different states in Nigeria (Ogun, n = 9; Kaduna, n = 6; Plateau, n = 3, and: Bauchi, n =

**Funding:** The author(s) received no specific funding for this work.

**Competing interests:** The authors have declared that no competing interests exist.

**Abbreviations:** AMR, Antimicrobial Resistance; DNA, Deoxyribonucleic acid; PFGE, Pulsed Field Gel Electrophoresis; EUCAST, European Union Committee on Antimicrobial Sensitivity Testing; UPGMA, Unweighted Pair Group Method with Arithmetic Means; CLSI, Clinical and Laboratory Standard Institute; PCR, Polymerase Chain Reaction; MAR, Multiple Antibiotic Resistance; MDR, Multi-Drug Resistance; ST198, Sequence Type 198; MIC, Minimum Inhibitory Concentration; QRDR, Quinolone Resistance Determining Region; PMQR, Plasmid Mediated Quinolone Resistance.

2). PFGE revealed 40 pulsotype patterns (Kentucky, n = 12; Larochelle, n = 9; Virchow, n = 5; Saintpaul, n = 4; Poona, n = 3; Isangi, n = 2, and; Nigeria, n = 2).

## Conclusion

This study recorded strictly related but diversely distributed *Salmonella* serovars with high AMR rates in poultry. We recommend strict regulation on antimicrobial use and regular monitoring of AMR trends among bacteria isolated from animals and humans to inform public policy.

## Introduction

Salmonellosis as a public health problem cannot be overemphasized. It remains one of the leading bacterial infections in humans, accounting for 93.8 million foodborne illnesses and over 155,000 annual deaths globally [1].

Most *Salmonella* infections are self-limiting, especially in humans; however, in the case of invasive infections or severe diarrhea, antimicrobial treatment is required. Fluoroquinolones and extended-spectrum β-lactams (particularly cephalosporins) are frequently the first-line antimicrobials used to treat salmonellosis. Fluoroquinolones have the potential to cause arthropathy and, hence, are not recommended in the treatment of infections in pediatric patients. However, fluoroquinolones are still considered one of the last treatment options for life-threatening *Salmonella* infections caused by multi-drug resistant (MDR) isolates [2]. The massive use of antimicrobial agents in livestock production as well as in humans favors the survival of MDR pathogens. Resistant *Salmonella* isolates recovered from different sources have been frequently reported worldwide [3, 4]. The increasing occurrence of MDR *Salmonella*, with resistance not only against the first-line antimicrobials, such as trimethoprim/sulphamethoxazole, chloramphenicol, and ampicillin, but also against WHO-listed critically important antimicrobials for human medicine, such as fluoroquinolones and third-generation cephalosporins (which have been extensively used in human and veterinary medicine), is a significant emerging public health concern [5, 6].

With the growing population and the increasing demand for animal protein in Nigeria, commercial farmers entering the poultry section are primed for higher profitability. In a bid to mitigate the impact of mortality associated with *Salmonella* on their farms, farmers resort to the use of antimicrobials without sound diagnostic advice, which results in the development of resistant isolates [7] and drug residues in the tissue of treated animals [8, 9]. Despite this practice, information about *Salmonella*'s genetic determinants of resistance to these antibiotics is still scanty in Nigeria.

Very little data is available concerning *Salmonella* serovars circulating in poultry production in Nigeria [10]. Moreover, there is a knowledge gap concerning the variability of the isolates within each serovar. This study was carried out (both from the phenotypic and genotypic points of view) to explore the spread of antimicrobial resistance (AMR) among *Salmonella* isolated from laying hen farms in Nigeria. Moreover, we investigated the similarities (epidemiological correlation) among the isolates. These data are essential to identify putative sources for defining effective control measures.

## Materials and methods

Two hundred *Salmonella* isolates from a previous study [10] were selected (based on the frequency, serovar, and geographical and sample type distribution) and analyzed to determine

their antimicrobial resistance profiles (see also the S1 File). PCR protocols were used to assess the presence of specific genetic determinants of resistance, while for isolates whose mode of resistance to certain antimicrobials (fluoroquinolones and olymyxins) was by gene mutation, gene sequence analysis was carried out. Pulsed-field gel electrophoresis (PFGE) was conducted on the seven most common serovars to determine the genetic relatedness among the isolates.

## Antimicrobial susceptibility profiling

The antimicrobial susceptibility patterns of the *Salmonella* isolates were determined using the European Union Committee on Antimicrobial Sensitivity Testing (EUCAST) minimum inhibitory concentration (MIC) method [11]. The isolates were tested by using Sensititre plates (TREK Diagnostic System, EUVSEC) against a panel of 14 different antimicrobials commonly used in veterinary and human medicine to treat bacterial infections, namely: sulfamethoxazole SMX (8–1024 µg/ml); tetracycline TET (2–64 µg/ml); azithromycin AZI (2–64 µg/ml); trimethoprim TMP (0.25–32 µg/ml); chloramphenicol CHL (8–128 µg/ml); meropenem MERO (0.03–16 µg/ml); ciprofloxacin CIP (0.015–8 µg/ml); nalidixic acid NAL (4–128 µg/ml); gentamicin GEN (0.5–32 µg/ml); ampicillin AMP (1–64 µg/ml); tigecycline TGC (0.25–8 µg/ml); colistin COL (1–16 µg/ml); cefotaxime FOT (0.25–4 µg/ml), and; ceftazidime TAZ (0.5–8 µg/ml).

The EUCAST epidemiological cut-offs were used (11), except for SMX and KAN, where the CLSI breakpoints were considered [12]. *Escherichia coli* ATCC 25922 served as the control strain in this study. We defined AMR as resistance to one or more classes of antimicrobials (there were 8 classes among the 14 antimicrobials) [13], whereas MDR isolates were defined as having resistance to three or more classes of antimicrobials. The formula: multiple antibiotic resistance (MAR) index = (number of antibiotics that an isolate is resistant to)/(total number of antibiotics tested) was calculated for all the isolates [14].

## Detection of antimicrobial resistance genes

According to the resistance profiles obtained, the isolates were analyzed to identify the responsible genetic determinants. Template DNA was prepared by boiling pure bacterial culture for 10 min. Both simplex and multiplex PCR analyses were carried out to detect the plasmid-mediated quinolone resistance (PMQR) genes (*qnrA*, *qnrB*, *qnrS*), tetracycline resistance genes (*tetA*, *tetB*), chloramphenicol acetyltransferase genes (*catA1*, *cmlA1*, *floR*), sulphonamide resistance genes (*sul1*, *sul2*, *sul3*), dihydrofolate reductase genes (*dfrA10*, *dfrA12*, *dfrA5-14*, *dfrA7-17*), and aminoglycoside modifying enzyme genes (*aac(3)-le*, *aacC4*, *aacC2*, *aadB*, *armA*) [15–17]. PCR was also carried out to amplify quinolone resistance determining region (QRDR), genes encoding for gyrase subunit A (*gyrA*), topoisomerase IV subunit C (*parC*) and the polymyxin reductase genes (*pmrA*, *pmrB*), as well as the temoxicillinase gene $bla_{TEM}$ [18].

## Sequencing of *gyrA*, *parC*, *pmrA* and *tem* genes

Genetic determinants were identified by DNA sequencing of the amplicons using a Genetic Analyzer 3130XL (Applied Biosystems, Life Technologies Corporation, Carlsbad, California). The sequences obtained were compared using the BLAST program (http://blast.ncbi.nlm.nih.gov/Blast.cgi) to determine point mutations.

## Pulsed field gel electrophoresis

For the PFGE analysis, 7 different serovars (*S.* Kentucky n = 35, *S.* Poona n = 15, *S.* Isangi n = 8, *S.* Saintpaul n = 9, *S.* Nigeria n = 5, *S.* Virchow n = 9, and *S.* Larochelle n = 13) were

selected based on the frequency of isolation of the serovar, antimicrobial susceptibility pattern, as well as the geographical and sample type distributions. All 94 selected *Salmonella* isolates were characterized by PFGE, which was carried out after the digestion of genomic DNA with the restriction enzyme *Xba*I [19]. Gel images were analyzed by BioNumerics version 7.6 (Applied Maths, Sint-Martens-Latem, Belgium) and compared by cluster analysis using the Dice coefficient for similarity and the unweighted pair group method with arithmetic means (UPGMA), with a position tolerance of 2% and optimization of 1%.

## Ethical considerations

We obtained written informed consent from farm owners before samples were collected from their farms. We guaranteed the respondents that the data we collected would be kept confidential. Permission was also obtained from the management of the national Veterinary Research Institute (NVRI) and the Federal Ministry of Agriculture and Rural Development. The Research Ethics Committee of NVRI gave the ethical approval for the project.

## Results

### Determination of antimicrobial resistance profiles

The MIC analysis of the 200 isolates showed that 97 (48.5%) were resistant to at least one antimicrobial agent. Five isolates of *S.* Isangi, three of *S.* Kentucky, two of *S.* Poona and one of *S.* Saintpaul were resistant to 7 (50%) of the antibiotics tested. Resistance to ciprofloxacin was common in all the MDR isolates (Table 1), while all the tested isolates were susceptible to cefotaxime, ceftazidime, and meropenem (Table 2). The MAR index ranged from 0.07–0.5 (Table 1).

The PCR and sequencing results of resistance genes showed the presence of targeted genes (Table 3). The detection of *pmrA*, *gyrA*, *parC*, *qnr*, *tem*, *catA1*, *cmlA1*, *floR*, *dfrA5-14*, *sul2*, *aac (3)-le*, and *tetA* genes as well as mutations in the nucleotides of *parC*, *gyrA*, and *pmrA* with subsequent amino acid substitution, as shown in Table 4, provides an insight to the AMR of these isolates (see also the S2–S4 Files).

All *S.* Kentucky isolates were resistant to at least three classes of antibiotics. In particular, the predominant resistance pattern, CIP/GEN/NA/SMX/TET, was observed in 73% (32/44) of the isolates. All *S.* Kentucky isolates showed resistance to fluoroquinolones and quinolones (ciprofloxacin and nalidixic acid), and this resistance was due to a double mutation of *gyrA* (Ser83 and Asp87) genes in 95% of the isolates (19/20). One isolate showed only one mutation in the *gyrA* (Ser83) gene and simultaneously harbored *qnrS* and *qnrA* plasmid genes. Only two isolates of *S.* Kentucky showed the presence of *qnrB* genes in addition to the chromosomal mutations occurring on the *gyrA* gene. The *tetA* and *aac(3)-le* genes were present in all 20 isolates. Most (95%) of the *S.* Kentucky isolates showed the *sul1* gene (19/20), while only one isolate was positive for the *sul2* gene. Eight isolates showed phenotypic resistance to ampicillin with the subsequent detection of the *bla*$_{TEM}$ gene. The TEM variants observed were TEM1, TEM57, and TEM30 (Table 3).

Two of the *S.* Poona isolates (2/6) were resistant to ampicillin, chloramphenicol, tetracycline, ciprofloxacin, tigecycline, and sulphomethoxazole. Resistance to ciprofloxacin was seen in most (4/6) of the *S.* Poona isolates, and this resistance was related to the presence of a mutation in the *parC* (Thr53) gene in addition to the presence of the plasmid-mediated resistance gene, *qnrB*. Four isolates showed phenotypic resistances to chloramphenicol, ampicillin, and sulphamethoxazole, with the subsequent detection of *cmlA1*, *tem57*, *and sul3* genes, respectively. One isolate showed a double mutation in *pmrA* (T89 and V93) for colistin resistance (Table 4).

**Table 1. Antimicrobial resistance profiles of *Salmonella* isolates from commercial layer farms in Nigeria.**

| Serial Number | Serovar | Antibiotic resistance pattern(s) | No. (%) of isolates | MAR index |
|---|---|---|---|---|
| 1 | *S.* Kentucky | CHL, CIP, NA | 1 (1.03) | 0.21 |
| | | CIP, GEN, NA, SMX, TET | 32 (32.99) | 0.36 |
| | | AMP, CIP, GEN, NA, SMX, TET, | 8 (8.25) | 0.43 |
| | | AMP, CHL, CIP, GEN, NA, SMX, TET | 3 (3.09) | 0.5 |
| 2 | *S.* Poona | CIP, NA | 1 (1.03) | 0.14 |
| | | GEN, TMP | 1 (1.03) | 0.14 |
| | | AMP, CHL, CIP, SMX | 1 (1.03) | 0.29 |
| | | AMP, CHL, SMX, TET, TGC | 1 (1.03) | 0.36 |
| | | AMP, CHL, CIP, COL, SMX, TET, TGC | 1 (1.03) | 0.5 |
| | | AMP, CHL, CIP, NA, SMX, TET, TGC | 1 (1.03) | 0.5 |
| 3 | *S.* Isangi | CIP, NA | 5 (5.15) | 0.14 |
| | | AZI, CHL, CIP, NA, SMX, TET, TMP | 5 (5.15) | 0.5 |
| 4 | *S.* Saintpaul | CHL, CIP, TET | 1 (1.03) | 0.21 |
| | | AMP, AZI, CHL, CIP, SMX, TET, TMP | 1 (1.03) | 0.5 |
| 5 | *S.* Nigeria | CIP, NA, TET, TGC | 2 (2.06) | 0.29 |
| 6 | *S.* Virchow | CIP, NA, SMX, TET, TMP | 7 (7.22) | 0.36 |
| 7 | *S.* Larochelle | CIP | 2 (2.06) | 0.07 |
| | | CIP, NA | 4 (4.12) | 0.14 |
| | | CIP, GEN, NA | 1 (1.03) | 0.21 |
| | | AMP, CHL, CIP, GEN, SMX | 1 (1.03) | 0.36 |
| 8 | *S.* Chomedey | CIP, SMX, TET | 2 (2.06) | 0.21 |
| | | CIP, NA, SMX, TET | 3 (3.09) | 0.29 |
| 9 | *S.* Telelkebir | CHL, CIP, SMX, TET | 2 (2.06) | 0.29 |
| 10 | *S.* Kingston | CIP, SMX, TET | 1 (1.03) | 0.21 |
| 11 | *S.* Schwarzengrund | SMX, TET | 3 (3.09) | 0.14 |
| 12 | *S.* Bonariensis | CHL, SMX | 1 (1.03) | 0.14 |
| 13 | *S.* Lomita | CIP, NA | 1 (1.03) | 0.14 |
| 14 | *S.* Muenster | CIP, NA | 1 (1.03) | 0.14 |
| 15 | *S.* Chichester | SMX | 1 (1.03) | 0.07 |
| 16 | *S.* Give | AZI | 1 (1.03) | 0.07 |
| 17 | *S.* Liverpool | COL | 1 (1.03) | 0.07 |
| 18 | *S.* Enteritidis | COL | 1 (1.03) | 0.07 |
| Total | | | 97 | |

All (10/10) of the *S.* Isangi isolates were resistant to the quinolones (ciprofloxacin and nalidixic acid) and in addition, five of the 10 isolates were resistant to azithromycin, chloramphenicol, sulphamethoxazole, tetracycline, and trimethoprim. All but one of the isolates had both *qnrB* and *qnrS* genes. One of the isolates had a mutation in *gyrA* (Leu55) in addition to having the plasmid genes, *qnrB* and *qnrS*. Resistance genes *sul2* (sulphomethoxazole), *tetA* (tetracycline), *dfrA5-14* (trimethoprim), and *catA1* (chloramphenicol) were detected in two of the isolates (Table 3).

The MIC analysis performed on the two *S.* Saintpaul isolates showed they were resistant to chloramphenicol, ciprofloxacin, and tetracycline. In addition, one of these isolates was also resistant to ampicillin, azithromycin, sulphamethoxazole, and trimethoprim. A mutation was observed in the *parC* (Thr57) gene in one of the isolates. Both isolates had the resistance genes *qnrS* and *tetA*. Resistance genes *tem1*, *sul2*, *dfrA5-14*, *catA1*, and *floR* were detected in one isolate. Similarly, the one *S.* Bonariensis isolate studied harbored resistance genes *sul3* and *cmlA1*.

**Table 2. Percentage (number of resistant/number tested) of antimicrobial resistance among the most common *Salmonella* serovars to 14 antibiotics used in *Salmonella* treatment.**

| Serovar | AMP | AZI | CHL | CIP | COL | GEN | NAL | SMX | TET | TGC | TMP | MERO[b] | FOT[c] | TAZ[d] | Total |
|---|---|---|---|---|---|---|---|---|---|---|---|---|---|---|---|
| *S.* Kentucky [a,] | 25 (11/44) | 0 (0/44) | 9.1 (4/44) | 100 (44/44) | 0 (0/44) | 97.7 (43/44) | 100 (44/44) | 97.7 (43/44) | 97.7 (43/44) | 0 (0/44) | 0 (0/44) | 0 (0/44) | 0 (0/44) | 0 (0/44) | 37.7 (232/616) |
| *S.* Poona | 67 (4/6) | 0 (0/6) | 67 (4/6) | 67 (4/6) | 17 (1/6) | 17 (1/6) | 33 (2/6) | 67 (4/6) | 50 (3/6) | 50 (3/6) | 17 (1/6) | 0 (0/6) | 0 (0/6) | 0 (0/6) | 32.1 (27/84) |
| *S.* Isangi | 20 (2/10) | 50 (5/10) | 50 (5/10) | 100 (10/10) | 0 (0/10) | 0 (0/10) | 100 (10/10) | 50 (5/10) | 50 (5/10) | 0 (0/10) | 50 (5/10) | 0 (0/10) | 0 (0/10) | 0 (0/10) | 33.6 (47/140) |
| *S.* Saintpaul | 50 (1/2) | 50 (1/2) | 100 (2/2) | 100 (2/2) | 0 (0/2) | 0 (0/2) | 0 (0/2) | 50 (1/2) | 100 (2/2) | 0 (0/2) | 50 (1/2) | 0 (0/2) | 0 (0/2) | 0 (0/2) | 35.7 (10/28) |
| *S.* Nigeria | 0 (0/2) | 0 (0/2) | 0 (0/2) | 100 (2/2) | 0 (0/2) | 0 (0/2) | 100 (2/2) | 0 (0/2) | 100 (2/2) | 100 (2/2) | 0 (0/2) | 0 (0/2) | 0 (0/2) | 0 (0/2) | 28.6 (8/28) |
| *S.* Virchow | 0(0/7) | 0(0/7) | 0(0/7) | 100 (7/7) | 0(0/7) | 0(0/7) | 100 (7/7) | 100 (7/7) | 100 (7/7) | 0(0/7) | 100 (7/7) | 0(0/7) | 0(0/7) | 0(0/7) | 35.7 (35/98) |
| *S.* Larochelle | 12.5 (1/8) | 0 (0/8) | 12.5 (1/8) | 100 (8/8) | 0 (0/8) | 25 (2/8) | 62.5 (5/8) | 0 (0/8) | 0 (0/8) | 0 (0/8) | 0 (0/8) | 0 (0/8) | 0 (0/8) | 0 (0/8) | 15.2 (17/112) |
| *S.* Chomedey | 0 (0/5) | 0 (0/5) | 0 (0/5) | 100 (5/5) | 0 (0/5) | 0 (0/5) | 60 (3/5) | 100 (5/5) | 100 (5/5) | 0 (0/5) | 0 (0/5) | 0 (0/5) | 0 (0/5) | 0 (0/5) | 25.7 (18/70) |
| *S.* Schwarzengrund | 0 (0/3) | 0 (0/3) | 0 (0/3) | 0 (0/3) | 0 (0/3) | 0 (0/3) | 0 (0/3) | 100 (3/3) | 100 (3/3) | 0 (0/3) | 0 (0/3) | 0 (0/3) | 0 (0/3) | 0 (0/3) | 14.3 (6/42) |
| *S.* Telelkebir | 0 (0/2) | 0 (0/2) | 100 (2/2) | 100 (2/2) | 0 (0/2) | 0 (0/2) | 0 (0/2) | 100 (2/2) | 100 (2/2) | 0 (0/2) | 0 (0/2) | 0 (0/2) | 0 (0/2) | 0 (0/2) | 28.6 (8/28) |
| *S.* Kingston | 0 (0/1) | 0 (0/1) | 0 (0/1) | 100 (1/1) | 0 (0/1) | 0 (0/1) | 0 (0/1) | 100 (1/1) | 100 (1/1) | 0 (0/1) | 0 (0/1) | 0 (0/1) | 0 (0/1) | 0 (0/1) | 21.4 (3/14) |
| *S.* Lomita | 0 (0/1) | 0 (0/1) | 0 (0/1) | 100 (1/1) | 0 (0/1) | 0 (0/1) | 100 (1/1) | 0 (0/1) | 0 (0/1) | 0 (0/1) | 0 (0/1) | 0 (0/1) | 0 (0/1) | 0 (0/1) | 14.3 (2/14) |
| *S.* Muenster | 0 (0/1) | 0 (0/1) | 0 (0/1) | 100 (1/1) | 0 (0/1) | 0 (0/1) | 100 (1/1) | 0 (0/1) | 0 (0/1) | 0 (0/1) | 0 (0/1) | 0 (0/1) | 0 (0/1) | 0 (0/1) | 14.3 (2/14) |
| *S.* Chichester | 0 (0/1) | 0 (0/1) | 0 (0/1) | 0 (0/1) | 0 (0/1) | 0 (0/1) | 0 (0/1) | 100 (1/1) | 0 (0/1) | 0 (0/1) | 0 (0/1) | 0 (0/1) | 0 (0/1) | 0 (0/1) | 7.1 (1/14) |
| *S.* Give | 0 (0/1) | 100 (1/1) | 0 (0/1) | 0 (0/1) | 0 (0/1) | 0 (0/1) | 0 (0/1) | 0 (0/1) | 0 (0/1) | 0 (0/1) | 0 (0/1) | 0 (0/1) | 0 (0/1) | 0 (0/1) | 7.1 (1/14 |
| *S.* Liverpool | 0 (0/1) | 0 (0/1) | 0 (0/1) | 0 (0/1) | 100 (1/1) | 0 (0/1) | 0 (0/1) | 0 (0/1) | 0 (0/1) | 0 (0/1) | 0 (0/1) | 0 (0/1) | 0 (0/1) | 0 (0/1) | 7.1 (1/14 |
| *S.* Enteritidis | 0 (0/1) | 0 (0/1) | 0 (0/1) | 0 (0/1) | 100 (1/1) | 0 (0/1) | 0 (0/1) | 0 (0/1) | 0 (0/1) | 0 (0/1) | 0 (0/1) | 0 (0/1) | 0 (0/1) | 0 (0/1) | 7.1 (1/14 |
| *S.* Bonariensis | 0 (0/1) | 0 (0/1) | 100 (1/1) | 0 (0/1) | 0 (0/1) | 0 (0/1) | 0 (0/1) | 100 (1/1) | 0 (0/1) | 0 (0/1) | 0 (0/1) | 0 (0/1) | 0 (0/1) | 0 (0/1) | 14.3 (2/14) |
| Total | 19.6 (19/97) | 7.2 (7/97) | 19.6 (19/97) | 89.7 (87/97) | 3.1 (3/97) | 47.4 (46/97) | 77.3 (75/97) | 75.3 (73/97) | 75.3 (73/97) | 5.2 (5/97) | 14.4 (14/97) | 0 (0/97) | 0 (0/97) | 0 (0/97) | 14.2 (189/1330) |

AMP = ampicillin: AZI = azithromycin: CHL = chloramphenicol: CIP = ciprofloxacin: COL = colistin: GEN = gentamicin: NAL = nalidixic acid:

SXM = sulphamethoxazole: TET = tetracycline: TGC = tigecycline: TMP = trimethoprim: MERO = meropenem: FOT = cefotaxime: TAZ = ceftazidime.

[a] = All the *S.* Kentucky isolates were susceptible to ciprofloxacin and nalidixic acid.

[b] = All the isolates were susceptible to meropenem.

[c] = All the isolates were susceptible to cefotaxime.

[d] = All the isolates were susceptible to ceftazidime.

The seven *S.* Virchow isolates had a resistance pattern of CIP/NA/SMX/TET/TMP, with mutation occurring in the *gyrA* gene (Ser83) in one of these isolates. Resistance genes *sul1*, *tetA*, and *dfrA5-14* were detected in two of these isolates. All the *S.* Chomedey isolates (n = 5) were resistant to ciprofloxacin, sulphamethoxazole, and tetracycline, with additional resistance to nalidixic acid in three of the isolates, which harbored a mutation occurring on the *parC* (Thr57) gene. All *S.* Chomedey isolates had resistance genes *sul2*, *tetA*, and *qnrB*.

**Table 3. Polymerase chain reaction (PCR) results showing the detected target genes of the most common *Salmonella* serovars isolated in Nigeria.**

| Serovar | AMP (*tem*) | SMX (*sul*) | TET (*tet*) | GEN (*aac(3)-le*) | TMP (dfrA) | CHL/FFN | (PMQR) *qnr* |
|---------|-------------|-------------|-------------|-------------------|------------|---------|--------------|
| Kentucky | *tem1; tem30; tem57* | *sul1; sul2* | *tetA* | *aac(3)-le* | | *cmlA1* | *qnrA; qnrB; qnrS* |
| Poona | *tem57* | *sul3* | *tetA* | - | - | *cmlA1* | *qnrB* |
| Isangi | - | *sul2* | *tetA* | - | *dfrA5-14* | *catA1* | *qnrB; qnrS* |
| Saintpaul | *tem1* | *sul2* | *tetA* | - | *dfrA5-14* | *catA1; floR* | *qnrS* |
| Nigeria | - | - | *tetA* | - | - | - | *qnrB* |
| Virchow | - | *sul1* | *tetA* | - | *dfrA5-14* | - | - |
| Larochelle | - | - | = | - | - | - | *qnrB* |
| Chomedey | - | *sul2* | *tetA* | - | - | - | *qnrB* |
| Lomita | - | - | - | - | - | - | *qnrB* |
| Bonariensis | - | *sul3* | - | - | - | *cmlA1* | - |
| Muenster | - | - | - | - | - | - | *qnrA* |

PCR SUL = Polymerase chain reaction for *sul* genes (sulphamethoxazole)

PCR GEN = Polymerase chain reaction for aac(3)-le gene (gentamicin)

PCR TMP = Polymerase chain reaction for *dfrA5-14* gene (trimethoprim)

PCR COL = Polymerase chain reaction for *pmrA* genes (colistin)

PCR AMP = Polymerase chain reaction for *tem gene* (ampicillin)

PCR TET = Polymerase chain reaction for *tet* genes (tetracycline)

Both *S.* Larochelle (n = 8) and *S.* Lomita (n = 1) had resistance only to fluoroquinolones. Resistance was related to the mutation occurring on the *parC* (Thr57) gene, in addition to the plasmid-mediated quinolone resistance gene, *qnrB*. *S.* Mueuster (n = 1) also had resistance only to fluoroquinolones; however, resistance was related to a mutation detected in the *gyrA* (Ser83) gene, in addition to the presence of the plasmid-mediated quinolone resistance gene, *qnrA*.

**Table 4. Sequence results showing the detected target genes and the amino acid substitutions (mutation) at different positions for the most common *Salmonella* serovars isolated in Nigeria.**

| Serovar | (QRDR) *gyrA* | (QRDR) *parC* | COL (*pmrA*) | *tem* genes |
|---------|---------------|---------------|--------------|-------------|
| Kentucky | S83→F; D87→G; S83→Y | - | - | G90 D (*tem57*); R241 S (*tem30*) |
| Poona | - | T57→S | T89→S; V93→G | G90 D (*tem57*) |
| Isangi | L55→F | - | - | |
| Saintpaul | - | T57→S | - | |
| Virchow | S83→Y | - | - | |
| Chomedey | - | T57→S | - | |
| Nigeria | - | T57→S | - | |
| Larochelle | - | T57→S | - | |
| Lomita | - | T57→S | - | |
| Bonariensis | - | - | - | |
| Muenster | S83→Y | - | - | |

S = serine; F = phenylalanine; L = leucine; D = aspartic acid; G = glycine; T = threonine; V = valine; Y = tyrosine; R = arginine

QRDR = quinolone resistance determining region

PMQR = plasmid mediated quinolone resistance

COL = colistin

## Comparison and cluster analysis of seven *Salmonella* serovars by PFGE

The PFGE results of *S.* Kentucky, *S.* Poona, *S.* Isangi, *S.* Saintpaul, *S.* Nigeria, *S.* Virchow, and *S.* Larochelle showed varying degrees of similarity/difference between the molecular weight (band) patterns of the restricted segments (pulsotypes). The comparison of PFGE profiles obtained from 94 isolates examined in this study identified PFGE cluster patterns for each of the seven serovars investigated, with significant ranges of genetic similarities (see also the S4 File).

*S.* Kentucky strains were classified into three clusters with different genetic similarities (Fig 1). Cluster 1 in particular, contained five out of 35 strains (three of them were indistinguishable, Dice coefficient = 100%), that were grouped into two different subgroups (Dice coefficient = 97.4%). The remaining 30 strains were grouped into two clusters with 10 different pulsotypes that were characterized by a high degree of genetic similarity (Dice coefficient ≥ 87.7%). Among the strains in cluster 2, 43% (15/35) of the total number of strains analyzed were indistinguishable (Dice coefficient = 100%). Most of the strains in each cluster were from different sample types and geographical locations.

*Salmonella* Poona strains were grouped into two main clusters with the intra-cluster strains probably being related (Dice coefficient = 85.1%). Each cluster contained pulsotypes with genetic similarity of 96% and 100%, meaning that they were strictly related genetically (in cluster 1) or indistinguishable (in cluster 2) (Fig 2). The strains belonging to cluster 2 were isolated from different sample types (water = 5, feed = 2, feces = 1, bootswab = 1, and dust = 1) and different geographical locations (states in Nigeria: Ogun and Lagos = Southwest, Kaduna = Northwest, Imo = Southeast). Cluster 1 also contained four indistinguishable strains isolated from different sample types and geographical locations.

*S.* Isangi isolates were classified into two different clusters, probably related (Dice coefficient = 82.8%). Each of the clusters had pulsotypes that were indistinguishable (Fig 3). Each cluster had strains that were from different sample types, but the same geographical locations (states in Nigeria: cluster 1 = Plateau and cluster 2 = Ogun).

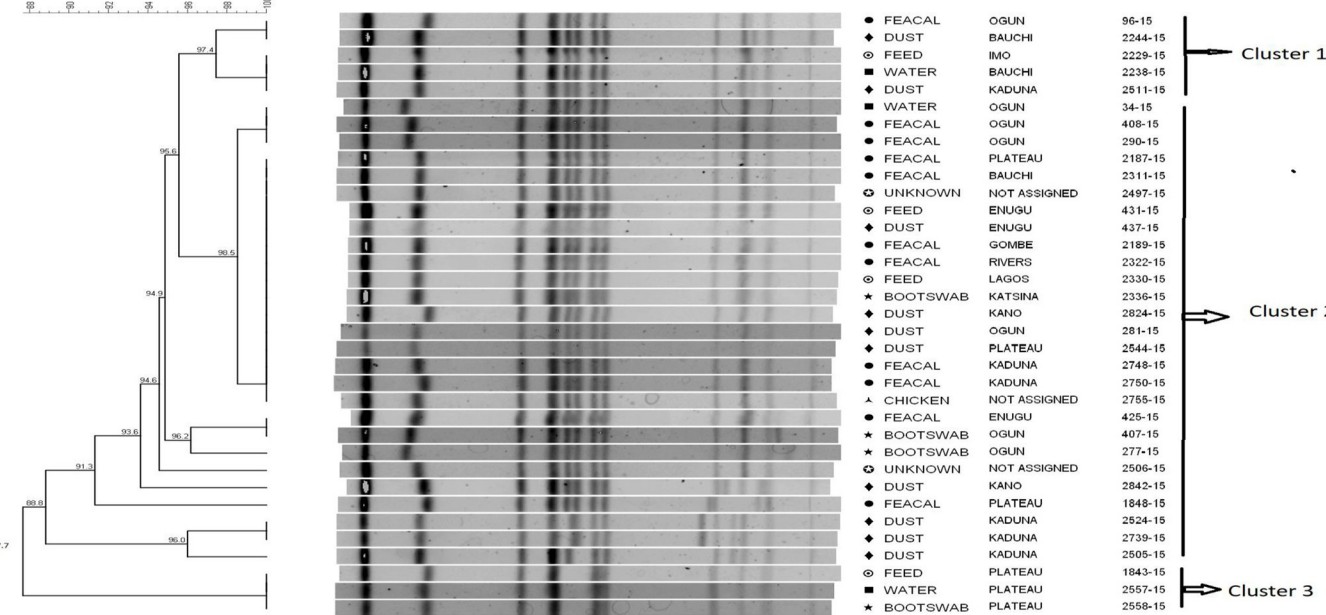

**Fig 1. Dendrogram showing the genetic similarity among the PFGE profiles of *Salmonella* Kentucky isolates.** The brackets highlight the main clusters detected (similarity of ≥ 88.8%) in the 35 isolates analyzed.

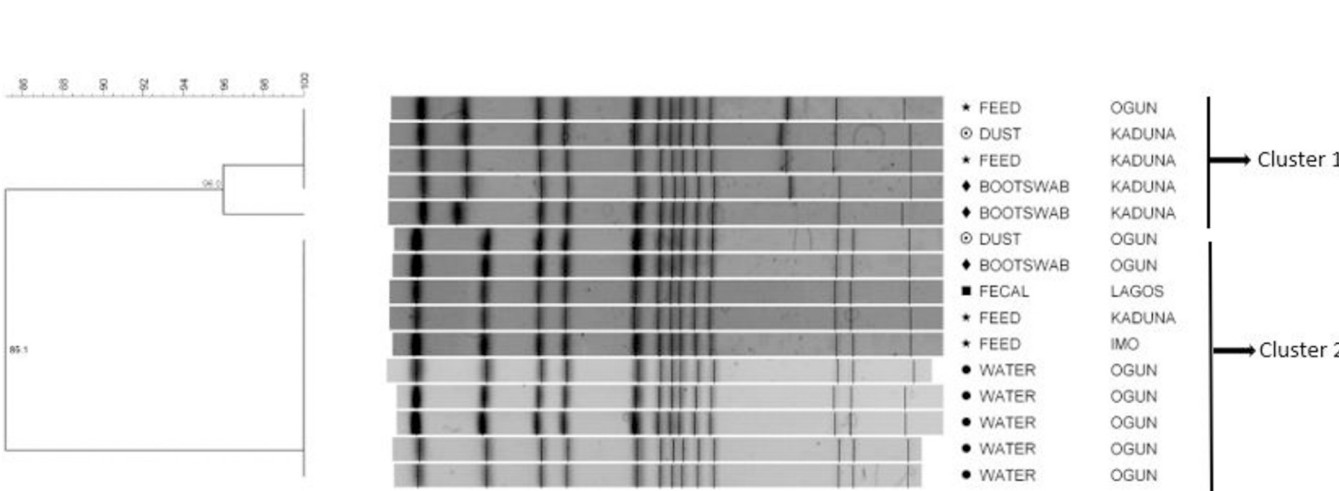

**Fig 2. Dendrogram showing the genetic similarity among the PFGE profiles of 15 *Salmonella* Poona isolates.** The brackets highlight the main clusters detected (similarity of 100%) in the 15 isolates analyzed.

*Salmonella* Saintpaul strains were classified into two main clusters, with the intra-cluster strains probably being related genetically (Dice coefficient = 82.1%). In the first cluster, each pulsotype was characterized by subgroups with strains that were genetically similar (89.7%), meaning that they were strictly related. Moreover, within each subgroup, there were indistinguishable strains (dice coefficient = 100%). The most prevalent pulsotype was shared by five of the nine *S.* Saintpaul strains investigated (Fig 4).

*Salmonella* Nigeria strains revealed two different clusters (Dice coefficient = 66.7%). Cluster 1 was characterized by four strains that were indistinguishable (Dice coefficient = 100%). Cluster 2 contained just one strain (Fig 5).

*Salmonella* Virchow strains were classified into three unrelated clusters (Dice coefficient = 63.3%). Cluster 1 had two pulsotypes with three subgroups that were strictly related (93.3% and 98.3%), and two of the subgroups were characterized by indistinguishable strains (Dice coefficient = 100%). Clusters 1 and 2 were characterized by strains that were probably related (Dice coefficient ≥ 78.9%). Cluster 3 had two isolates that were indistinguishable (Fig 6).

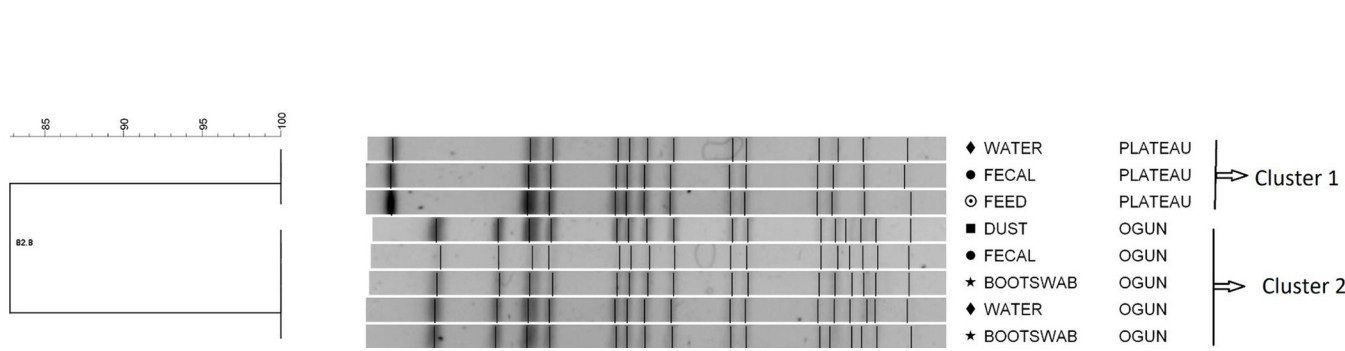

**Fig 3. Dendrogram showing the genetic similarity among the PFGE profiles of 8 *Salmonella* Isangi isolates.** The brackets highlight the main clusters detected (similarity of 100%) in the 8 isolates analyzed.

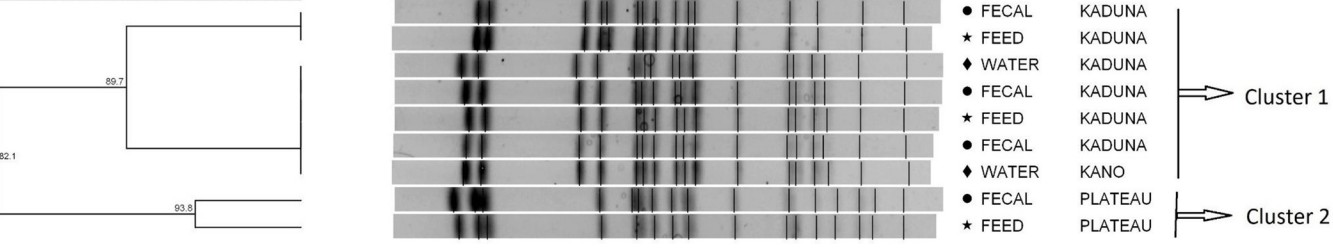

**Fig 4. Dendrogram showing the genetic similarity among the PFGE profiles of 9 *Salmonella* Saintpaul isolates.** The brackets highlight the main clusters detected that are probably related. (However, within cluster 1, there were 5 isolates that were indistinguishable).

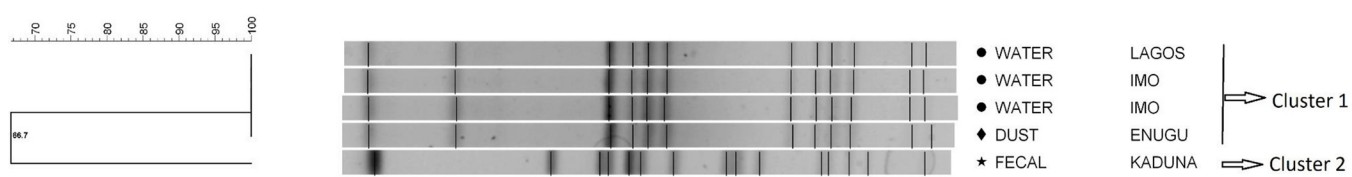

**Fig 5. Dendrogram showing the genetic similarity among the PFGE profiles of 5 *Salmonella* Nigeria isolates.** The brackets highlight the main clusters detected (similarity of 100% in 4 of the isolates) in the 5 isolates analyzed.

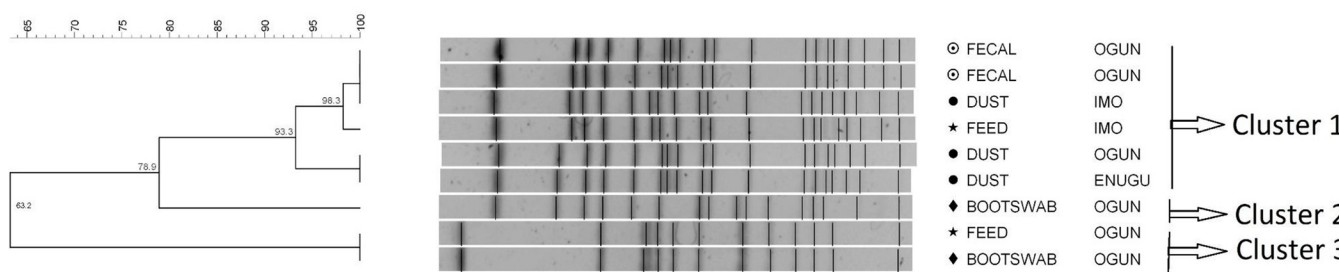

**Fig 6. Dendrogram showing the genetic similarity among the PFGE profiles of 9 unrelated *Salmonella* Virchow isolates.** The brackets highlight the main clusters, one of which had two pulsotypes with three subgroups that were strictly related (93.3% and 98.3%).

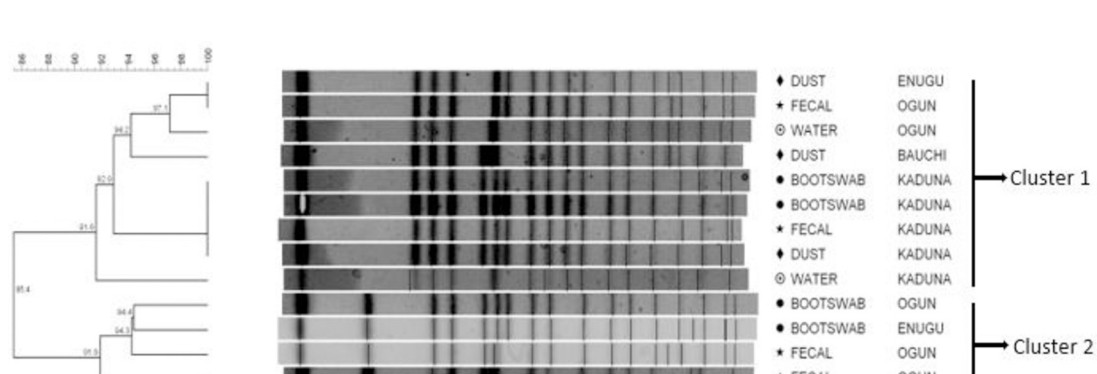

**Fig 7. Dendrogram showing the genetic similarity among the PFGE profiles of 13 *Salmonella* Larochelle isolates.** The brackets highlight the main clusters detected (similarity of 91.6% and 91.9, respectively) in the 13 isolates analyzed.

*Salmonella* Larochelle was classified into two main clusters that are probably related (Dice coefficient = 85.4%) (Fig 7). The two clusters contained different pulsotypes with genetic similarity of 91.6% to 91.9%, respectively, for clusters 1 and 2, meaning that the intra-cluster pulsotypes were strictly related genetically.

## Discussion

High rates of resistance to ciprofloxacin, nalidixic acid, sulphamethoxazole, tetracycline, and gentamicin were found in this study. Resistance rates to chloramphenicol, ampicillin, trimethoprim, azithromycin, tigecycline, and colistin were low. A comparative study in Ibadan, Nigeria, reported a high frequency of resistance to ciprofloxacin, tetracycline, nalidixic acid, and sulphamethoxazole among chicken isolates, supporting our findings [20]. Another study, [21], reported 100% resistance to fluoroquinolones for clinical isolates from chickens in the northern part of Nigeria, while [22] reported only one isolate from a chicken that showed ciprofloxacin resistance out of the thirty investigated isolates. Twelve isolates (*S.* Poona n = 2; *S.* Saintpaul n = 2; *S.* Larochelle n = 3; *S.* Chomedey n = 2; *S.* Telekebir n = 2, and; *S.* Kingston n = 1) in this study showed low susceptibility to ciprofloxacin and high susceptibility to nalidixic acid as was earlier reported by Fashae et al. [20].

All the *S.* Kentucky isolates were resistant to ciprofloxacin and nalidixic acid, while most were resistant to tetracycline and sulphamethoxazole. The multi-drug resistance displayed by *Salmonella* isolates in this study could suggest the emergence of co-resistance and/or cross-resistance. For example, low levels of resistance to colistin were seen, and this may be due to cross-resistance from ciprofloxacin, tetracycline, and sulphonamide resistances [23, 24]. Co-resistance to fluoroquinolones and third-and fourth-generation cephalosporins has also been identified in *Salmonella* isolated from humans [25].

Raufu *et al.* [7] reported significant contamination of poultry with multidrug-resistant *S.* Kentucky in Nigeria. The association of *S.* Kentucky with poultry dates back to 1937 in the United States, where it was isolated from chickens. Although multilocus sequence typing (MLST) was not carried out in this study, the patterns of resistance to antimicrobials we tested reaffirmed the initial speculation of a circulating, poultry-associated STI98-X1 CIP *S.* Kentucky clone in Africa, including in Nigeria [26].

The lack of policy in Nigeria to control the use of antimicrobials in poultry, especially fluoroquinolones, including ciprofloxacin, enrofloxacin, and ofloxacin [27], could have contributed to the rapid spread of AMR in poultry farms. In *Salmonella*, the acquisition of resistance genes likely occurred by conjugation, usually with other *Enterobacteriaceae*, or through the transfer of plasmids [28]. Bacteria that develop resistance via extended-spectrum β-lactamase genes could become a reservoir of resistance genes, which can enter the food chain [29]. Screening of resistance genes has proved to be a useful tool in the epidemiological tracing of genes between bacterial species, prediction of future outbreaks in different regions, and treatment and control strategies of bacterial diseases [30].

Resistance to gentamicin involves enzymatic modification of aminoglycosides [31]. These are acetylation, phosphorylation, and adenylation of hydroxyl and amine groups. Although aminoglycoside modifying enzymes are the leading cause of resistance, plasmid-mediated 16S rRNA-methylases (16S-RMTases) have been increasingly reported in recent years [32]. However, 16S-RMT-ases were not investigated in this work. There are scarce reports on aminoglycoside resistance genes in *Salmonella* species [30, 33], but we detected the gene for aminoglycoside modifying enzyme, *aac(3)-le*, in less than half of the isolates screened for gentamicin resistance. There is a similarity between the aminoglycoside gene detected in this study with that from *E. coli* in China [34]. Aminoglycoside-resistant isolates from foods of animal origin are of public health importance in developing countries because this class of antimicrobial is used to treat a wide variety of infections [35].

Tetracycline-resistance genes were the most frequently occurring resistance genes in this study, and all of the tetracycline-resistant isolates yielded targeted amplicons with *tetA* primers, but there were no amplification products detected by *tetB* and *tetG* primers. In this study, both tetracycline-susceptible and -resistant isolates harbored the tetracycline-resistance genes. These present results agree with those reported by both Deekshit *et al* [36] and Maynard *et al* [37], who showed that some antimicrobial-resistance genes of bacteria can be silent when resistance testing is carried out *in-vitro*. These silent genes, under the selection pressure of antibiotic use, can be transferred to other bacteria or be turned on *in-vivo* [38]. All *Salmonella* isolates that were resistant to sulphamethoxazole carried either the *sul1*, *sul2*, or *sul3* genes, with *sul1* being the most predominant. Most isolates resistant to sulphamethoxazole also showed resistance to ciprofloxacin, nalidixic acid, and tetracycline. The *dfrA5-14* gene was found in isolates that were resistant to trimethoprim. The *catA1*, *cmlA1*, and *floR* genes were detected among the chloramphenicol-resistant isolates tested. In Nigeria, chloramphenicol was previously the antibiotic of first choice for the treatment of *Salmonella*-related illnesses; however, a report found that chloramphenicol resistance increased by 72.4–89.2% between 1997 and 2007, limiting this drug's therapeutic usefulness [22].

Sequencing of the $bla_{TEM}$ genes revealed three variants. These were TEM 57 (substitution of amino acid G with D at position 90), TEM 30 (substitution of amino acid R with S at position 241), and TEM 1 (the reference sequence). Bae *et al*. [39] reported that ampicillin- and/or cephalosporin-resistant *Salmonella* strains can carry the $bla_{TEM}$ gene, with or without the $bla_{CTX-M}$ or $bla_{OXA}$ gene. However, only the $bla_{TEM}$ gene was detected in this study. This further confirms the low rates of resistance by these *Salmonella* isolates to the extended-spectrum β-lactam antibiotics, especially the cephalosporins. Mutations were observed in the *pmrA* gene (T89→S and V93→G) for colistin resistance. This amino acid substitution in the *pmrA* gene has also been reported by Hong and Ko [40].

Resistance to fluoroquinolones typically arises as a result of modifications in the target enzymes (DNA gyrase and topoisomerase IV), changes in drug entry/efflux, and plasmid-mediated quinolone (*qnr*) resistance, which protects the quinolone target proteins from inhibition [41]. Genes regulating the influx and efflux of quinolones into/from cells were not

investigated in this study. The genetic mutation in fluoroquinolone resistance is a step-wise phenomenon, affecting both *gyrA* and *parC* genes [42]. Most investigators have reported various substitutions of amino acid at different positions in the *gyrA* gene [43], but the substitution in this study, which was Leu55→Phe in the *gyrA* subunit, is extremely rare. Contrary to the report of Wang *et al*. [44], the mutation at position 87 has no comparative advantage for ciprofloxacin resistance over the mutation at position 83. Previous studies have not been able to detect *gyrB* or *parE* genes [42, 45]. Although *gyrB* and *parE* gene mutations were not detected in this study either, we did detect particular substitutions in g*yrA* and *parC*, which are considered major quinolone resistance mechanisms [42, 45].

Different plasmid-mediated quinolone resistance (PMQR) genes (*qnrA*, *qnrB*, *qnrC*, *qnrD*, *qnrS*, *qepA*, *aac(60)-Ib-cr*) have been described [46], but only *qnrA*, *qnrB*, and *qnrS* genes were detected in this study. While *S*. Kentucky, *S*. Saintpaul, and *S*. Isangi harbored the *qnrS* gene, only *S*. Kentucky and *S*. Muenchen had the *qnrA* gene. The PCR and sequencing results showed that plasmid-mediated quinolone resistance (*qnr*) genes were responsible for the resistances seen in four of the isolates that were resistant to ciprofloxacin but susceptible to nalidixic acid, even though two of the isolates had a single mutation in the *parC* gene.

In previous studies, researchers have demonstrated the relationship between the increased prevalence of antimicrobial-resistant bacteria and (i) the increased use of antimicrobials in human and veterinary medicine, (ii) greater movement of people and animals, and (iii) increased industrialization [47]. Ogun State, located in the southern part of Nigeria has a large poultry industry. This area not only meets the protein needs of its people through the production of day-old chicks and poultry products, but it also supplies other areas and imports animal feeds and ingredients from other regions [48]. This movement is a potential contributor to the spread of antibiotic-resistant bacteria [49]. The environment itself also contains a variety of bacteria that potentially present an immense pool of AMR genes, which if transferred between bacteria, could cause human and animal disease [50]. The high level of resistance observed in this study reaffirms the great importance of strengthening collaboration between veterinary and public health sectors on appropriate detection and reporting of zoonotic foodborne pathogens [51]. Despite bans and legislation, some antibiotics are still fed to livestock prophylactically and routinely to increase profits and to limit potential bacterial infections in stressed and crowded livestock [52]. Antimicrobial-resistant bacteria will inevitably follow wherever antimicrobials are used; therefore, a coordinated, multi-disciplinary approach will be required to address this issue [53]. Monitoring AMR trends among bacteria isolated from food, animals, and humans is necessary to inform public policy regarding the appropriate use of antimicrobial agents in veterinary and human medicine [54].

Many studies have reported the successful use of PFGE for assessing the genetic diversity of *Salmonella* isolates [55] and the clonal transmission of isolates in a country, a food industry, or a herd [56–58]. Veterinarians, food producers, and risk managers can utilize the PFGE-based database to trace the sources of contamination on farms and in food processing facilities. In the present study, fewer than half of the *S*. Kentucky isolates were indistinguishable or were possibly epidemiologically related, but they were isolated from four different states. PFGE has been successfully used for the sub-typing of several *Salmonella* serovars and other bacterial pathogens [59–61] as well as in the context of foodborne disease investigations. The results of the present study provided a baseline molecular PFGE database of the most prevalent *Salmonella* serovars in poultry in Nigeria, upon which other research works can be built.

Some *Salmonella* serovars are known to be associated with specific sources [62]. *S*. Derby, *S*. Montevideo and *S*. Kottbus have often been linked to pig, cattle, and poultry production systems, respectively [63]. Other serovars may also be linked to various sources. There was no specific association between most of the PFGE pulsotypes in this study and the sample types.

For example, all 20 indistinguishable *S*. Kentucky isolates were from the five different sample types analyzed.

## Limitation

This study is not without limitations, as it was impossible to purchase primers for some resistance genes. It was also not possible for us to carry out multilocus sequence typing (MLST) to confirm whether the *S*. Kentucky isolates were the multi-drug resistant ST198 XI strains that are of public health concern.

## Conclusion

This study showed high resistance rates of *Salmonella* isolates to fluroquinolones, with subsequent detection of relevant resistance genes especially amongst the *S*. Kentucky isolates. Resistances to other commonly used antibiotics in animal production were also detected. This could be attributed to the unregulated sales and use of antibiotics in the poultry industries in Nigeria. Although the *Salmonella* isolates were obtained from farms that were far from each other and collected during different time frames, a high level of genetic similarity was evidenced, demonstrating the presence of strictly related clones within the Nigerian layer farms. Based on our findings, we recommended that the antibiotic sensitivity of *Salmonella* isolates should be routinely verified by clinicians to determine each isolate's susceptibilities before prescribing or administrating antibiotics. Moreover, antimicrobials cannot be considered the solution to control *Salmonella* in poultry flocks. Some other alternatives, such as the application of biosecurity measures and vaccination of animals, must be considered, as they more effectively limit the spread of *Salmonella* (Regulation EC N° 1177/2006). There is a need for regulatory authorities to monitor AMR trends among bacteria isolated from food, animals, and humans, which will inform public policy regarding the appropriate use of antimicrobial agents in veterinary and human medicine. The high rate of antibiotic resistance in *Salmonella* observed in this study reaffirms the great importance of strengthening collaboration between veterinary and public health sectors on appropriate detection and reporting of zoonotic foodborne pathogens.

## Supporting information

**S1 File. Extracted tables from Fagbamila *et al*., 2017.**
(DOCX)

**S2 File. Polymerase chain reaction (PCR) procedures.**
(DOCX)

**S3 File. Sequence results showing amino acid substitution of different resistant genes.**
(DOCX)

**S4 File. PCR and PFGE gel images.**
(DOCX)

## Acknowledgments

The authors would like to appreciate the federal government of Nigeria through the competitive agricultural research grant scheme (CARGS) and the International Centre for Genetic Engineering and Biotechnology (ICGEB) for their support towards the success of the investigation.

## Author Contributions

**Conceptualization:** Idowu Oluwabunmi Fagbamila, Maryam Muhammad, Jacob K. P. Kwaga, Jarleth U. Umoh, Junaidu A. Kabir, Lisa Barco.

**Data curation:** Idowu Oluwabunmi Fagbamila, Elena Ramon, Maryam Muhammad.

**Formal analysis:** Idowu Oluwabunmi Fagbamila, Antonia A. Lettini, Lisa Barco.

**Funding acquisition:** Idowu Oluwabunmi Fagbamila, Maryam Muhammad.

**Investigation:** Idowu Oluwabunmi Fagbamila, Elena Ramon, Alessandra Longo, Keti Antonello.

**Methodology:** Idowu Oluwabunmi Fagbamila, Elena Ramon, Alessandra Longo, Keti Antonello, Mabel K. Aworh.

**Project administration:** Idowu Oluwabunmi Fagbamila, Maryam Muhammad, Mabel K. Aworh.

**Resources:** Maryam Muhammad, Paul A. Abdu, Antonia Ricci.

**Software:** Maryam Muhammad, Antonia Ricci.

**Supervision:** Maryam Muhammad, Antonia Ricci.

**Validation:** Antonia A. Lettini, Antonia Ricci, Lisa Barco.

**Writing – original draft:** Idowu Oluwabunmi Fagbamila, Mabel K. Aworh, Lisa Barco.

**Writing – review & editing:** Jacob K. P. Kwaga, Paul A. Abdu, Jarleth U. Umoh, Junaidu A. Kabir.

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
