## [Decision Letter · Decision Letter 0]

29 Jun 2023

PONE-D-23-07992ASSESSING THE MECHANISMS OF MULTI-DRUG RESISTANT NON-TYPHOIDAL SALMONELLA (NTS) SEROVARS FROM LAYER CHICKEN FARMS IN NIGERIAPLOS ONE

Dear Dr. Fagbamila,

Thank you for submitting your manuscript to PLOS ONE. After careful consideration, we feel that it has merit but does not fully meet PLOS ONE’s publication criteria as it currently stands. Therefore, we invite you to submit a revised version of the manuscript that addresses the points raised during the review process.

We look forward to receiving your revised manuscript.

Kind regards,

Nabi Jomehzadeh, Ph.D (Assistant Professor)

Academic Editor

PLOS ONE

Journal Requirements:

https://gutpathogens.biomedcentral.com/articles/10.1186/s13099-015-0063-3

https://www.frontiersin.org/articles/10.3389/fmicb.2015.00602/full

https://www.liebertpub.com/doi/10.1089/fpd.2007.0090

In your revision ensure you cite all your sources (including your own works), and quote or rephrase any duplicated text outside the methods section. Further consideration is dependent on these concerns being addressed.

The research was funded by the Federal government through the competitive agricultural research grant scheme (CARGS).

Additional Editor Comments:

We have received the reports from our advisors on your manuscript, "ASSESSING THE MECHANISMS OF MULTI-DRUG RESISTANT NON-TYPHOIDAL SALMONELLA (NTS) SEROVARS FROM LAYER CHICKEN FARMS IN NIGERIA," submitted to "PloS One."

Based on the advice received, I feel your manuscript could be reconsidered for publication should you be prepared to incorporate changes, as suggested by reviewers.

Reviewers' comments:

Reviewer's Responses to Questions

**Comments to the Author**

1. Is the manuscript technically sound, and do the data support the conclusions?

Reviewer #1: Yes

Reviewer #2: Yes

2. Has the statistical analysis been performed appropriately and rigorously? 

Reviewer #1: Yes

Reviewer #2: Yes

3. Have the authors made all data underlying the findings in their manuscript fully available?

Reviewer #1: Yes

Reviewer #2: Yes

4. Is the manuscript presented in an intelligible fashion and written in standard English?

Reviewer #1: Yes

Reviewer #2: Yes

5. Review Comments to the Author

Reviewer #1: A paragraph on the Salmonella spp. as a public health importance is suggested to be incorporated in the introduction section. There are some anomalies in reference writing in the text. It is suggested to follow the journal style properly.

Reviewer #2: Authors of the manuscript (PONE-D-23-07992) determines the MDR profiles of 200 NTS serovars and determine the mechanisms of resistance by different approaches. The research design and methodologies are acceptable and the data presentation and manuscript write-up are very good. The research findings/results might have significant values in determining AMR control strategies. Authors may consider few points to improve the manuscript further.

Minor Comments

1. Title of the manuscript may be rephrased as below to make it more meaningful:

ASSESSING THE MECHANISMS OF MULTI-DRUG RESISTANCE IN NON-TYPHOIDAL SALMONELLA (NTS) SEROVARS ISOLATED FROM LAYER CHICKEN FARMS IN NIGERIA

2. Authors used 200 isolates of NTS serovars from their previous study and determined their MDR profiles and mechanisms of MDR. Though they referred to their previous work and mentioned the criteria they followed to select 200 isolates, it might be good to include a table (might be a supplementary table) to describe the isolates based on the frequency, serovars, geographical and sample type distribution to help the readers.

3. In conclusion section, authors may describe their key findings shortly within few syntaxes followed by their concluding remarks/ recommendations.

6. PLOS authors have the option to publish the peer review history of their article (what does this mean?). If published, this will include your full peer review and any attached files.

Reviewer #1: No

Reviewer #2: **Yes: **Dr. Md. Masudur Rahman

---

## [Author Response · Author response to Decision Letter 0]

9 Aug 2023

Response to Editor comments

Response: The manuscript has been reviewed accordingly.

2. We noticed you have some minor occurrence of overlapping text with the following previous publication(s), which needs to be addressed: https://gutpathogens.biomedcentral.com/articles/10.1186/s13099-015-0063-3

https://www.frontiersin.org/articles/10.3389/fmicb.2015.00602/full

https://www.liebertpub.com/doi/10.1089/fpd.2007.0090

In your revision ensure you cite all your sources (including your own works), and quote or rephrase any duplicated text outside the methods section. Further consideration is dependent on these concerns being addressed.

Response: This has been corrected and can be seen in the manuscript with track changes

The name of the colleague or the details of the professional service that edited your manuscriptA copy of your manuscript showing your changes by either highlighting them or using track changes (uploaded as a *supporting information* file)

Response: This has been corrected and can be seen in the manuscript with track changes. A clean copy of the edited manuscript has also been attached. The colleagues that edited the manuscript are Dr. Mabel Ahwor and Lisa Barco

The research was funded by the Federal government through the competitive agricultural research grant scheme (CARGS).

Response: This section has been removed and rightly placed under acknowledgement

Response: A section on ethical consideration inserted under the Method section.

Response: Gel images of both the PCR and PFGE has been placed in the supplementary file

Response: Reference section has been reviewed and corrected using Mendeley

Response to reviewers’ comment

Reviewer 1

Reviewer #1: A paragraph on the Salmonella spp. as a public health importance is suggested to be incorporated in the introduction section. 

Response: A paragraph on the public health importance of Salmonella has been added.

2. There are some anomalies in reference writing in the text. It is suggested to follow the journal style properly.

Response: Reference section has been reviewed and corrected using Mendeley

Reviewer 2

Title of the manuscript may be rephrased as below to make it more meaningful:

ASSESSING THE MECHANISMS OF MULTI-DRUG RESISTANCE IN NON-TYPHOIDAL SALMONELLA (NTS) SEROVARS ISOLATED FROM LAYER CHICKEN FARMS IN NIGERIA

Response: Correction effected as suggested by reviewer. i.e the word ISOLATED was inserted after SEROVARS

Authors used 200 isolates of NTS serovars from their previous study and determined their MDR profiles and mechanisms of MDR. Though they referred to their previous work and mentioned the criteria they followed to select 200 isolates, it might be good to include a table (might be a supplementary table) to describe the isolates based on the frequency, serovars, geographical and sample type distribution to help the readers.

Response: I have attached extracted tables from previous paper (Fagbamila et al., 2017) as a supplementary document to guide the readers (See also the supplementary files S1).

In conclusion section, authors may describe their key findings shortly within few syntaxes followed by their concluding remarks/ recommendations.

Response: A sentence on the Key findings has been inserted in the first paragraph of the ‘Conclusion Section’

---

## [Editor Report · Decision Letter 1]

15 Aug 2023

ASSESSING THE MECHANISMS OF MULTI-DRUG RESISTANT NON-TYPHOIDAL SALMONELLA (NTS) SEROVARS ISOLATED FROM LAYER CHICKEN FARMS IN NIGERIA

PONE-D-23-07992R1

Dear Dr. Fagbamila,

We’re pleased to inform you that your manuscript has been judged scientifically suitable for publication and will be formally accepted for publication once it meets all outstanding technical requirements.

Kind regards,

Nabi Jomehzadeh, Ph.D (Assistant Professor)

Academic Editor

PLOS ONE

Additional Editor Comments (optional):

After reviewing the responses to the reviewers' comments, the amendments were accepted
---

## [Editor Report · Acceptance letter]

21 Aug 2023

PONE-D-23-07992R1 

ASSESSING THE MECHANISMS OF MULTI-DRUG RESISTANT NON-TYPHOIDAL *SALMONELLA* (NTS) SEROVARS ISOLATED FROM LAYER CHICKEN FARMS IN NIGERIA 

Dear Dr. Fagbamila:

I'm pleased to inform you that your manuscript has been deemed suitable for publication in PLOS ONE. Congratulations! Your manuscript is now with our production department. 

Kind regards, 

on behalf of

Dr. Nabi Jomehzadeh 

Academic Editor

PLOS ONE